# Costs of home-delivered antiretroviral therapy refills for persons living with HIV: Evidence from a pilot randomized controlled trial in KwaZulu-Natal, South Africa

**Ashley S. Tseng** [1,2]*, **Ruanne V. Barnabas** [3,4], **Alastair van Heerden** [5,6], **Xolani Ntinga** [5], **Maitreyi Sahu** [7]

1 Department of Epidemiology, University of Washington, Seattle, Washington, United States of America, 2 Department of Global Health, University of Washington, Seattle, Washington, United States of America, 3 Division of Infectious Diseases, Department of Medicine, Massachusetts General Hospital, Harvard Medical School, Boston, Massachusetts, United States of America, 4 Department of Epidemiology, Harvard T.H. Chan School of Public Health, Boston, Massachusetts, United States of America, 5 Center for Community Based Research, Human Sciences Research Council, Sweetwaters, KwaZulu-Natal, South Africa, 6 South African Medical Research Council/Wits Developmental Pathways for Health Research Unit, Department of Paediatrics, School of Clinical Medicine, Faculty of Health Sciences, University of the Witwatersrand, Johannesburg, Gauteng, South Africa, 7 Department of Health Metrics Sciences, University of Washington, Seattle, Washington, United States of America

* ashleystseng@gmail.com

⊘ OPEN ACCESS

**Data Availability Statement:** The Deliver Health Study human research participant data cannot be shared publicly because of participant locator

## Abstract

Antiretroviral therapy (ART) is needed across the lifetime to maintain viral suppression for people living with HIV. In South Africa, obstacles to reliable access to ART persist and are magnified in rural areas, where HIV services are also typically costlier to deliver. A recent pilot randomized study (the Deliver Health Study) found that home-delivered ART refills, provided at a low user fee, effectively overcame logistical barriers to access and improved clinical outcomes in rural South Africa. In the present costing study using the provider perspective, we conducted retrospective activity-based micro-costing of home-delivered ART within the Deliver Health Study and when provided at-scale (in a rural setting), and compared to facility-based costs using provincial expenditure data (covering both rural and urban settings). Within the context of the pilot Deliver Health Study which had an average of three deliveries per day for three days a week, home-delivered ART cost (in 2022 USD) $794 in the first year and $714 for subsequent years per client after subtracting client fees, compared with $167 per client in provincial clinic-based care. We estimated that home-delivered ART can reasonably be scaled up to 12 home deliveries per day for five days per week in the rural setting. When delivered at-scale, home-delivered ART cost $267 in the first year and $183 for subsequent years per client. Average costs of home delivery further decreased when increasing the duration of refills from three-months to six- and 12-month scripts (from $183 to $177 and $135 per client, respectively). Personnel costs were the largest cost for home-delivered refills while ART drug costs were the largest cost of clinic-based refills. When provided at-scale, home-delivered ART in a rural setting not only offers clinical

information. Data are available from Mr. Torin Schaafsma (ttss@uw.edu) at the University of Washington International Clinical Research Center for researchers who meet the criteria for access to confidential data. Ethical approval was obtained from the Human Sciences Research Council Research Ethics Committee in South Africa (REC 1/21/11/18) and the University of Washington Institutional Review Board in Seattle, Washington, United States (STUDY00005739).

**Funding:** The Deliver Health Study was funded by the U.S. National Institute of Mental Health (NIMH) (R21MH115770) and awarded to RVB and AVH. AST received support through an administrative diversity supplement from NIMH (R01MH124465-02S1). The funders had no role in study design, data collection and analysis, decision to publish, or preparation of the manuscript.

**Competing interests:** RVB declares support from the U.S. National Institutes of Health, and the Bill and Melinda Gates Foundation. Regeneron Pharmaceuticals covered the cost of abstract and manuscript writing outside the submitted work. RVB serves on a Gilead Sciences Data Monitoring Committee for which she receives an honorarium. AVH declares support from the U.S. National Institutes of Health, and the Bill and Melinda Gates Foundation. Outside the submitted work, MS declares payments to institution from the National Pharmaceutical Council and consulting fees for data analysis used by the Pharmaceutical Care Management Association. For the remaining authors none were declared. The funders had no role in study design, data collection and analysis, decision to publish, or preparation of the manuscript.

benefits for a hard-to-reach population but is also comparable in cost to the provincial standard of care.

## Introduction

Until a cure for HIV is available, reliable access to antiretroviral therapy (ART) is needed across the lifetime to maintain viral suppression. South Africa has one of the highest burdens of HIV worldwide with an HIV prevalence of 16% among adults aged 15 and older and a total of 7.8 million people with HIV who require lifetime ART [1]. In the past two decades, the South African government has invested heavily in their HIV programme, and has made great progress towards achieving the UNAIDS 95-95-95 treatment targets: as of 2022, 90% of South African adults with HIV knew their status, 91% of adults who knew their status were on ART, and 94% of people on ART were virally suppressed [1]. However, the status of HIV control varies considerably across the nation, with KwaZulu-Natal province continuing to have the highest HIV prevalence in the country (22%) [1] and rural farming communities reporting higher HIV prevalence than urban areas (18% vs. 13%) [2]. Patients attending clinics in rural South Africa have reported greater ART availability and affordability barriers than those in urban areas [3], and ART uptake in rural KwaZulu-Natal was found to be strongly negatively associated with distance from the nearest primary healthcare facility [4]. Access to HIV care is further constrained by slowing government funding which has not matched the level of demand [5], particularly in rural areas where the cost of achieving HIV viral suppression is higher due to logistical barriers, supply chain considerations, and staffing challenges [6]. In light of these rural/urban disparities in HIV outcomes and costs, novel and low-cost strategies to address gaps in HIV care in rural South Africa are critically needed.

A number of strategies have been proposed to improve HIV outcomes and reduce the burden of care associated with one to three-month ART refill visits for both clients and health systems [7]. In particular, the World Health Organization now recommends Differentiated Service Delivery (DSD) for HIV care, which involves tailoring programs to accommodate client needs to improve HIV treatment outcomes and reduce health system burden [8]. Various client-centered DSD mechanisms are being proposed to increase viral suppression by overcoming barriers to accessing HIV care including long wait times at clinics, transportation barriers, and social stigma [9–11]. As a result, alternative ART dispensing mechanisms which reduce clinic burden including home delivery of medications are being explored and have had demonstrated feasibility and success in improving client adherence in South Africa [7, 12–14].

The recent Deliver Health Study found that home-delivered ART and monitoring for a small user fee increased viral suppression by 21% compared to standard facility-based care among 162 participants in rural KwaZulu-Natal [15]; however, the costs of this program have not previously been quantified. The objective of the present study was to estimate the cost of home-delivered ART for adults living with HIV in rural South Africa (1) within the context of the pilot study and (2) when provided at-scale, compared to standard clinic-based ART resupply. A secondary objective was to identify cost drivers for each mode of ART delivery. In a sensitivity analysis, we also estimated cost savings when home-delivered ART used multi-month refill scripts. This study aims to inform the South African National Department of Health's (NDoH) strategy to improve HIV treatment outcomes and reduce health system burden in rural South Africa.

## Materials and methods

### Deliver Health Study

The Deliver Health Study (NIH R21MH115770; ClinicalTrials.gov Identifier: NCT04027153) was a pilot randomized trial of home ART delivery, monitoring, and ART resupply conducted in three rural and peri-urban communities in uMgungundlovu District, KwaZulu-Natal from October 2019 to December 2020. The detailed study procedures have been previously published [16], and are also summarized in the S1 File flow map. Briefly, 162 ART-eligible [17] adults aged 18 or over with HIV were identified through home and mobile van HIV testing and counselling and at HIV clinics. Following the enrollment and ART initiation visit, clients were randomized to receive either home delivery of HIV care which was provided for a small user fee of 30–90 South African Rand (ZAR) (2–7 United States Dollars [USD]), or standard clinic-based ART follow-up care at a health center of choice within the study catchment area. Home delivery of HIV care was provided leveraging an optimized delivery algorithm to reduce travel times and associated costs, described further in the S2 File. For the home delivery arm of the pilot study, study staff made deliveries three days per week, spent an average of 26 minutes per home delivery of ART (including transportation time), and made an average of three deliveries a day. Home delivery was at first implemented by a team of three (nurse, driver, data collector), but staffing was later reduced to just the nurse due to pandemic social distancing restrictions (S1 File). All Deliver Health Study participants received the TDF/FTC/EFV ART regimen for the treatment of HIV.

### Ethics statement and declarations

Ethical approval was obtained from the Human Sciences Research Council Research Ethics Committee in South Africa (REC 1/21/11/18) and the University of Washington Institutional Review Board in Seattle, Washington, United States (STUDY00005739). All study participants provided written informed consent per local ethics committee requirements. Access to de-identified, anonymized Deliver Health Study data was originally granted on March 8, 2022 to AST for reporting of participant demographics statistics.

### Study population

**Deliver Health Study participant demographics.** A total of 155 adults living with HIV completed follow-up in the Deliver Health Study, 81 (52%) of whom received home-delivered ART refills and monitoring and 74 (48%) of whom received clinic-based care. The median age of participants was 38 years (interquartile range: 12), and 72 (47%) were women. Seventy-one (88%) people living with HIV in the home-delivery group were virally suppressed at study exit compared to 55 (74%) people living with HIV in the clinic-based care group. The average monthly salary of participants randomized to the home delivery group, self-reported at baseline, was 1376 ZAR (108 USD) per individual with a standard deviation of 2247 ZAR (176 USD).

**Study setting.** KwaZulu-Natal was the province in South Africa with the highest estimated HIV prevalence of 22% among adults aged 15 years and older in 2022 [1], totaling approximately 1.9 million adults [18]. An estimated 52% of the population in KwaZulu-Natal resided in rural areas [19], and rural districts of KwaZulu-Natal had the highest estimated HIV prevalence in South Africa of greater than 28% [20]. At the end of 2022, 31% of 15–64 year olds residing in KwaZulu-Natal were unemployed [21] and the average monthly earnings of employees in the formal non-agricultural sector was 26,032 ZAR across South Africa [22].

## Cost estimation

Our primary analysis compared the cost of quarterly home delivery of ART (the pilot Deliver Health Study intervention) with the standard of care (SOC), i.e., clinic-based care with three-month refill scripts, in KwaZulu-Natal province. Due to limited capacity for costing during the trial (which was conducted during the COVID-19 pandemic), costs were collected retrospectively in December 2023–March 2024 for the period of October 2019–December 2020, and the home-delivery and SOC costs relied on different data sources representing different regions of KwaZulu-Natal, described further below. All costs were estimated from the perspective of the South African NDoH (provider) and represented the costs of a program implemented in the public sector, to inform the South African NDoH's financial planning and strategy. To quantify resource use [23], we first outlined program activities in a detailed Narrative Summary for the Deliver Health Study (S1 File) which further informed our assumptions regarding programmatic implementation (S2 File). Our primary outcomes of interest were:

1. Average annual cost per client: the average annual cost per person living with HIV in the Deliver Health Study or in KwaZulu-Natal

2. Average annual cost per client virally suppressed: the average annual cost per person living with HIV who was virally suppressed in the Deliver Health Study or in KwaZulu-Natal.

In our secondary objective, costs for each mode of service delivery were disaggregated by cost categories (personnel, vehicles and fuel, drugs, etc.) to understand the main cost drivers (detailed methods in the S2 File).

Costs were reported in 2022 USD. For costs originally reported in ZAR, we used the World Bank gross domestic product implicit deflator for South Africa [24] and the average ZAR-to-USD exchange rate in 2022 [25]. We deflated costs reported in USD using the US annual average Consumer Price Index [26]. Capital items and other costs with a useful life of more than one year were discounted 3%. Full details of these adjustments are provided in the S2 File.

**Costing of home delivery of ART and monitoring.**   For home-delivered refills, we conducted activity-based micro-costing for home-delivered ART and monitoring in the Deliver Health Study in rural uMgungundlovu District of KwaZulu-Natal. We excluded costs for research-specific procedures (e.g., data collection, research-related personnel costs). Building rental costs for storage of ART stock and supplies, and equipment costs for ART monitoring for clients were gathered from the implementing institution, the Human Sciences Research Council. Personnel salaries were obtained from the Deliver Health Study and South Africa Department of Public Service and Administration [27]. To estimate personnel time, vehicle costs, and fuel, we used recorded travel logs for the home delivery trips. We assumed the home-delivery intervention was delivered by a team of a nurse, a driver, and a community outreach worker. Full cost assumptions including staff time and vehicle costs are detailed in the S2 File.

**Costing of clinic-based ART refills and care.**   To estimate costs for clinic-based care, we used provincial government expenditure data for the full province of KwaZulu-Natal and scaled it to ART-specific care. We obtained actual HIV/AIDS spending (i.e., the rectified budget) for financial years 2019 and 2020 from the South African National Treasury [28]. We then estimated the proportion of the HIV program budget used for ART using a 2020 government HIV/AIDS spending report [29] and applied this proportion to the total 12-month HIV program budget. Full details for these calculations are in the S2 File. Of note, clinic expenditure data covered both rural as well as urban areas in KwaZulu-Natal (e.g., the major cities of Durban and Newcastle), where ART costs may be lower than in rural regions [6].

## Scenarios

For the home-delivered ART intervention, we modelled two sets of scenarios over a 12-month time horizon: programmatic and at-scale. Our baseline estimation assumed three-month (quarterly) refills; in a sensitivity analysis we also estimated the costs of home delivery for six- and 12-month ART refill scripts starting after the first quarter. Costs of home-delivered ART for all scenarios were compared to the SOC, i.e., clinic-based care with three-month ART refill scripts. Full details of the estimation for all scenarios are in the S2 File.

**Programmatic cost estimation.** The primary cost was the programmatic scenario for home-delivered ART, which we estimated to reflect the resource requirements if home-delivered ART were implemented by the South African NDoH. Accordingly, we substituted staff salaries paid under the Deliver Health Study with corresponding salaries reported by the South African government [27]. The programmatic scenario assumed that ART was delivered by a team of two (nurse and driver) who spent approximately 3.5 hours a day for 3 days a week making home deliveries to support 81 clients in the Deliver Health Study, and that the program was also supported by a community outreach worker and other operational staff (S2 File). In a sensitivity analysis, we also estimated a second programmatic "as-observed" scenario using staff salaries from within the Deliver Health Study.

**At-scale cost estimation.** Given the low client volume within the pilot Deliver Health Study, we estimated a scenario where home delivery of ART was scaled up to accommodate a larger client volume. In the Deliver Health Study, the maximum number of deliveries made in a given day was 16 visits and the average time per visit was 26 minutes (including driving). With this information and assuming 214 workdays per year (excluding weekends, holidays, and other leave), five workdays per week, and eight-hour workdays, we estimated that the same team could reasonably make 12 home deliveries per day, spending approximately six hours of their workday making home deliveries to clients and about two hours to load and unload the delivery vehicle at the central office. In addition, we assumed an ART initiation visit (which includes HIV counselling) would take three times as long as an ART refill visit, which was qualitatively informed by time-and-motion observations conducted during January–February 2023 for an ongoing follow-up trial to the Deliver Health Study (S2 File).

## Results

### Programmatic scenarios

For the 81 clients receiving home-delivered ART in the Deliver Health Study, cost of home delivery was substantially higher than provincial estimates for annual per-client facility-based ART costs. Specifically, we estimated that in the context of the pilot study the average annual cost per client receiving home-delivered ART refills was $794 per client and $907 per client virally suppressed in the first year of intervention and $714 per client and $815 per client virally suppressed for subsequent years, after subtracting paid client fees (Tables 1 and 2). In comparison, the SOC was estimated to cost an average of $167 annually per client and $254 per client virally suppressed. We found that "as-observed" costs for home-delivered ART in the Deliver Health Study were slightly higher than "as-implemented" costs in the primary programmatic scenario due to hiring costs in the study ($819 per client and $935 per client virally suppressed for quarterly refills in the first year of intervention; see Supplementary Table 3.2.1 in S3 File). Multi-month scripting scenarios (with six- and 12-monthly refills) further reduced the cost of home-delivered ART, as shown in Table 2.

For home-delivered ART, the largest cost categories were personnel costs (33%), buildings and overhead (17%), and vehicles and fuel (11%) for the first year of implementation (Fig 1A).

**Table 1. Average annual cost per client and average annual cost per virally suppressed client by refill method within the context of the Deliver Health Study, under the programmatic scenario with 3-month ART refill scripts.**

| Cost category | Average annual cost per client | | | | | | Average annual cost per client virally suppressed | | | | | |
|---|---|---|---|---|---|---|---|---|---|---|---|---|
| | Home delivery[a] | | | | Clinic[c,d] N = 1,925,698 | | Home delivery[e] | | | | Clinic[c,d] N = 1,270,961 | |
| | First year[b] N = 81 | | Subsequent years N = 81 | | | | First year[b] N = 71 | | Subsequent years N = 71 | | | |
| | 2022 ZAR | 2022 USD | 2022 ZAR | 2022 USD | 2022 ZAR | 2022 USD | 2022 ZAR | 2022 USD | 2022 ZAR | 2022 USD | 2022 ZAR | 2022 USD |
| | Cost (% of total cost) | | | | | | | | | | | |
| ART drugs | 1,150 (9) | 70 (9) | 1150 (10) | 70 (10) | 1150 (42) | 70 (42) | 1,312 (9) | 80 (9) | 1,312 (10) | 80 (10) | 1,743 (42) | 106 (42) |
| Buildings and administrative overhead | 2,189 (17) | 134 (17) | 2,091 (18) | 128 (18) | 58 (2) | 4 (2) | 2,497 (17) | 152 (17) | 2,386 (18) | 146 (18) | 88 (2) | 5 (2) |
| Communication | 1,269 (10) | 78 (10) | 1,269 (11) | 78 (11) | 6 (0.22) | 0.36 (0.22) | 1,448 (10) | 88 (10) | 1,448 (11) | 88 (11) | 9 (0.22) | 0.55 (0.22) |
| Equipment | 1,200 (9) | 73 (9) | 150 (1) | 9 (1) | 2 (0.06) | 0.11 (0.06) | 1,369 (9) | 84 (9) | 172 (1) | 10 (1) | 3 (0.06) | 0.16 (0.06) |
| Hiring and training | 657 (5) | 40 (5) | 657 (6) | 40 (6) | 2 (0.09) | 0.15 (0.09) | 749 (5) | 46 (5) | 749 (6) | 46 (6) | 4 (0.09) | 0.22 (0.09) |
| Materials and supplies | 753 (6) | 46 (6) | 590 (5) | 36 (5) | 561 (20) | 34 (20) | 859 (6) | 52 (6) | 674 (5) | 41 (5) | 850 (20) | 52 (20) |
| Personnel wages and benefits | 4,354 (33) | 266 (33) | 4,354 (37) | 266 (37) | 960 (35) | 59 (35) | 4,968 (33) | 303 (33) | 4,968 (37) | 303 (37) | 1454 (35) | 89 (35) |
| Vehicles and fuel | 1,491 (11) | 91 (11) | 1,489 (13) | 91 (13) | 0.26 (0.01) | 0.02 (0.01) | 1,701 (11) | 104 (11) | 1,699 (13) | 104 (13) | 0.40 (0.01) | 0.02 (0.01) |
| Total[f] | 13,006 | 794 | 11,694 | 714 | 2,740 | 167 | 14,846 | 907 | 13,349 | 815 | 4,151 | 254 |

Abbreviations: ART = antiretroviral therapy, ZAR = South African Rand, USD = United States Dollars, HIV = Human Immunodeficiency Virus, UNAIDS = Joint United Nations Programme on HIV/AIDS.

[a]There were 81 people living with HIV on ART in the home-delivered ART refills group of the Deliver Health Study.

[b]Includes startup costs.

[c]The standard of care in South Africa is 3-month ART refills at clinics.

[d]The total number of adults (aged 15 years and older) living with HIV in KwaZulu-Natal in September 2022 was 1,925,698 people. Source: UNAIDS HIV sub-national estimates viewer.

[e]There were 71 people living with HIV on ART who were virally suppressed at study exit in the home-delivered ART refills group of the Deliver Health Study.

[f]The average fee for home delivery service, about 4 USD, paid by clients in the home delivery intervention of the Deliver Health Study was subtracted from the programmatic costs of implementing home delivery.

These proportions remained relatively stable for subsequent years (Fig 1B). For the SOC, the largest costs were ART drugs (42%), personnel costs (35%), and materials and supplies (20%) for each year of implementation (Fig 1A). Total annual costs by cost category are reported in Supplementary Table 3.3.1 in S3 File.

## At-scale scenarios

We estimated that if the home delivery team was working full-time (using the parameters described in the methods), the team could feasibly make 12 home delivery stops per working day and accommodate a client volume of 367 newly diagnosed people with HIV in the first year and up to 642 clients in subsequent years. We estimated that this at-scale programmatic cost of home delivery would be $267 annually per client for home-delivered ART refills and monitoring in the first year and $183 in subsequent years (Table 3), in comparison to $167 for clinic-based refills. The average annual cost per client virally suppressed receiving home-

**Table 2. Estimated cost implications of multi-month home-delivered and standard clinic-based ART refills and monitoring as-observed by the South African National Department of Health with South African government salaries, under the programmatic scenario with 3-, 6-, and 12-month ART refill scripts.**

| Scenario | Average annual cost per client | | | | | | Average annual cost per client virally suppressed | | | | | |
|---|---|---|---|---|---|---|---|---|---|---|---|---|
| | Home delivery | | | | Clinic[b] N = 1,925,698 | | Home delivery | | | | Clinic[b] N = 1,270,961 | |
| | First year[a] N = 81 | | Subsequent years N = 81 | | | | First year[a] N = 71 | | Subsequent years N = 71 | | | |
| | 2022 ZAR | 2022 USD | 2022 ZAR | 2022 USD | 2022 ZAR | 2022 USD | 2022 ZAR | 2022 USD | 2022 ZAR | 2022 USD | 2022 ZAR | 2022 USD |
| Programmatic costs with 3-month refills | 13,006 | 794 | 11,694 | 714 | 2,740 | 167 | 14,846 | 907 | 13,349 | 815 | 4,151 | 254 |
| Programmatic costs with 6-month refills | 10,014 | 612 | 9,337 | 570 | | | 11,424 | 698 | 10,652 | 651 | | |
| Programmatic costs with 12-month refills | 8,261 | 504 | 7,521 | 459 | | | 9,424 | 576 | 8,581 | 524 | | |

Abbreviations: ART = antiretroviral therapy, ZAR = South African Rand, USD = United States Dollars, HIV = Human Immunodeficiency Virus, UNAIDS = Joint United Nations Programme on HIV/AIDS.

[a]Includes startup costs.

[b]The standard of care in South Africa is 3-month ART refills at clinics.

delivered ART refills would be $303 in the first year and $208 in subsequent years (Table 3) vs. $254 for clients receiving clinic-based care.

Multi-month scripting further reduced the cost of home-delivered ART so that it was comparable or even lower in cost than SOC. With six-month ART refill scripts, the annual cost per client would be $225 for home-delivered refills in the first year and $177 for subsequent years, (compared to $167 for the SOC), and the average cost per client virally suppressed would be $255 in the first year and $201 in subsequent years (vs. $254 for the SOC). At-scale 12-month ART refills would be cost-saving after the first year: the annual cost per client would be $135 in subsequent years (compared to $167 for the SOC) (Table 3).

For three-month at-scale home delivered ART, the largest first-year costs were ART drugs (26%), personnel costs (22%), materials and supplies (17%), and buildings and overhead (12%) (Fig 2A). The rankings of cost categories remained the same for subsequent years (Fig 2B). Average client costs per cost category for the six-month refill programmatic and at-scale scenarios are reported in the S3 File, along with additional cost driver figures.

## Discussion

In this costing study of home-delivered ART refills in a high-prevalence rural setting in South Africa, we estimate that when delivered at-scale, home delivery costs in a rural setting are comparable to the SOC of clinic-based refills administered across both rural and urban settings. Specifically, assuming a reasonable client volume of 12 home deliveries per day, we estimate that the annual per-client cost of home-delivered ART is $267 in the first year and $183 in subsequent years, as compared to the province-level average of $167 for clinic-based refills; this corresponds to a per-client incremental cost of $100 in the first year and $16 in subsequent years (Table 3). Multi-month scripting (six- and 12-months) even further reduced costs of home delivery compared with standard clinic-based refills, and were either comparable (six-months) or cost-saving (12-months) when provided at-scale after the first year (Table 3). The published Deliver Health Study results demonstrated clinical effectiveness of home-delivered ART [16], and the present costing study suggests that home-delivered ART can be feasibly scaled up in a high prevalence rural setting in South Africa at modest incremental cost.

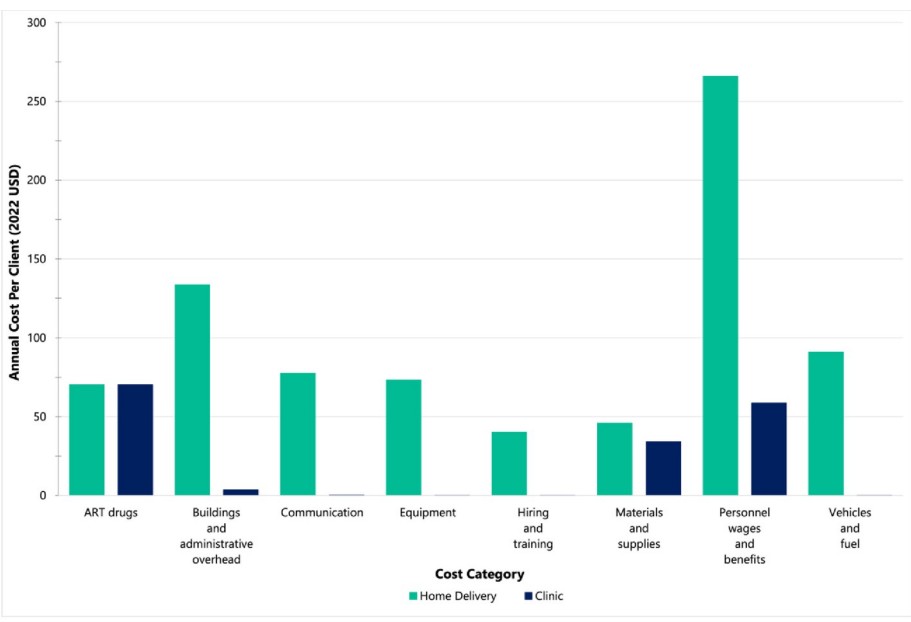

A

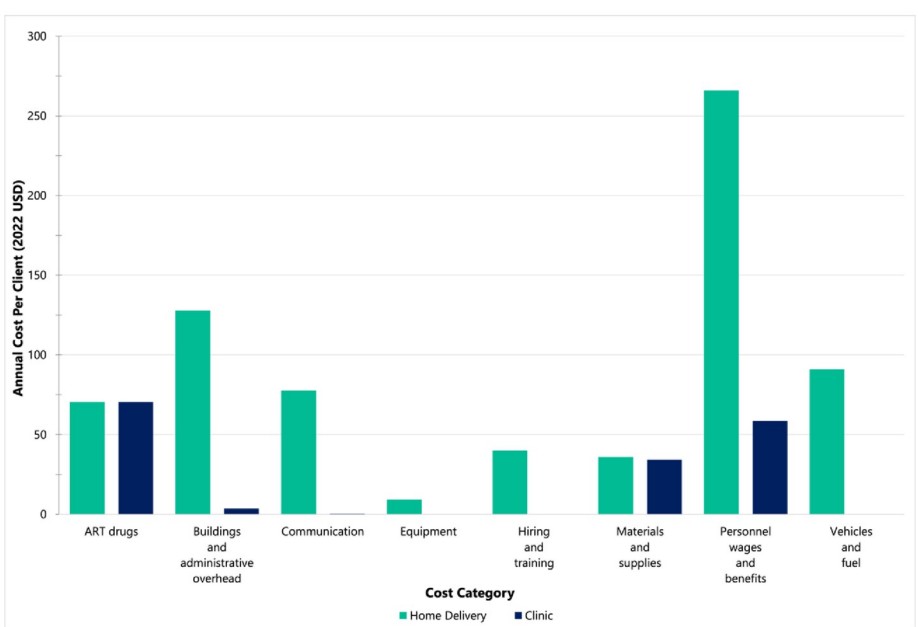

B

**Fig 1. Average annual cost per client (2022 USD) for home-delivered 3-month ART refills vs. clinic-based 3-month ART refills (standard of care) by cost category in the programmatic NDoH-implemented scenario.** The NDoH scenario assumes fixed costs as implemented in the Deliver Health Study and public sector clinical staff salaries instead of study salaries. (A) Home-delivered 3-month ART refills (first year costs) vs. standard of care. (B) Home-delivered 3-month ART refills (subsequent year costs) vs. standard of care.

Interestingly, the largest cost category for home-delivered ART was personnel wages and benefits (33% in the baseline scenario) and not vehicle and fuel costs (11%) (Fig 1). Personnel was the largest cost for a number of reasons. First, within the Deliver Health Study, staff

**Table 3. Estimated cost implications of multi-month home-delivered and standard clinic-based ART refills and monitoring provided at-scale by the South African National Department of Health with South African government salaries, with 3-, 6-, and 12-month ART refill scripts.**

| Scenario | Average annual cost per client | | | | | | Average annual cost per client virally suppressed | | | | | |
|---|---|---|---|---|---|---|---|---|---|---|---|---|
| | Home delivery | | | | Clinic N = 1,925,698 | | Home delivery | | | | Clinic N = 1,270,961 | |
| | First year[b] N = 367 | | Subsequent years N = 642 | | | | First year[b] N = 323 | | Subsequent years N = 565 | | | |
| | 2022 ZAR | 2022 USD | 2022 ZAR | 2022 USD | 2022 ZAR | 2022 USD | 2022 ZAR | 2022 USD | 2022 ZAR | 2022 USD | 2022 ZAR | 2022 USD |
| At-scale programmatic costs with 3-month refills[a] | 4,367 | 267 | 2,996 | 183 | 2,740 | 167 | 4,963 | 303 | 3,405 | 208 | 4,151 | 254 |
| At-scale programmatic costs with 6-month refills | 3,681 | 225 | 2,900 | 177 | | | 4,183 | 255 | 3,296 | 201 | | |
| At-scale programmatic costs with 12-month refills | 3,921 | 239 | 2,214 | 135 | | | 4,455 | 272 | 2,516 | 154 | | |

Abbreviations: ART = antiretroviral therapy, ZAR = South African Rand, USD = United States Dollars, HIV = Human Immunodeficiency Virus, UNAIDS = Joint United Nations Programme on HIV/AIDS.

[a]The standard of care in South Africa is 3-month ART refills.

[b]Includes startup costs.

coordinated and made individual home deliveries based on client delivery preferences. In addition, the protocol required that clients were physically present at the time of delivery (vs. dropping off refills at their front door). Thus, planning of delivery trips and routes driven were not always optimized. With a greater volume of clients, optimized delivery algorithms adapted from routing science-guided delivery in the private sector [30–32] could be useful in managing client delivery preferences and delivery personnel time, and may ultimately lower programmatic costs. This point is evidenced by our findings in the at-scale scenarios, in which ART drugs contributed the largest costs to the home-delivery intervention in both three- and six-month refill scenarios. Second, we assumed that programmatic implementation of home-delivered ART included a full-time professional nurse at grade 2 and a driver. These programmatic nurse salaries were in fact higher than the nurse staff salaries from the Deliver Health Study, which is generally not consistent with the literature [33, 34]. Alternative team configurations with task shifting to lower cadre health workers (e.g., lower grade nurses or community health workers) could be tested in future work to determine effectiveness and efficiency. Finally, vehicles did not contribute a substantial portion of total estimated costs due to low diesel and vehicle costs relative to personnel costs.

In the Deliver Health Study, clients paid a one-time fee, tiered based on individuals' income (US $2, $4, or $6), for a six-month ART home delivery service (which was extended due to COVID-19). User fees in general were well-accepted: in the Deliver Health Study, 98% of clients paid the full user fee for the home-delivery ART service and 100% reported willingness to continue to pay a fee for the service due to its perceived convenience and flexibility [11]. Notably, these client fees for the home delivery service did not substantially offset delivery costs (personnel, vehicle, and fuel) in either the as-observed or at-scale scenarios. Moreover, the fee in the Deliver Health Study covered home ART delivery service for six months, thus an annual service fee per client would likely be higher (e.g., double that of the six-month service) in implementation. However, the user fees still contributed to a modest reduction in programmatic costs and may additionally have contributed to program effectiveness [35, 36].

Our estimates somewhat differ from other costing studies for DSD and ART delivery in South Africa. First, this comparison covers a rural setting for home-delivered ART and a combined rural/urban setting for the SOC–the difference in setting may contribute to an increase

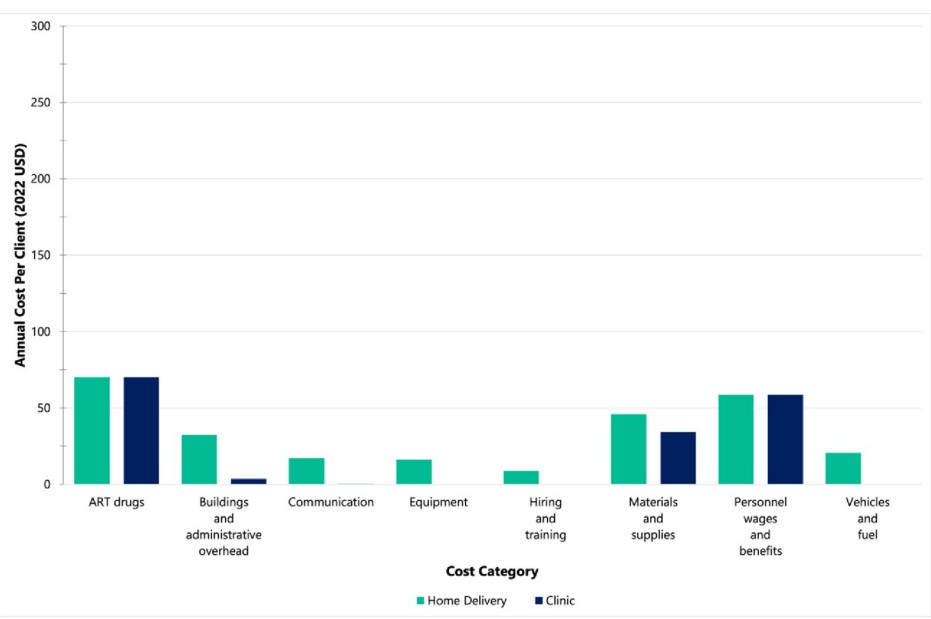

**A**

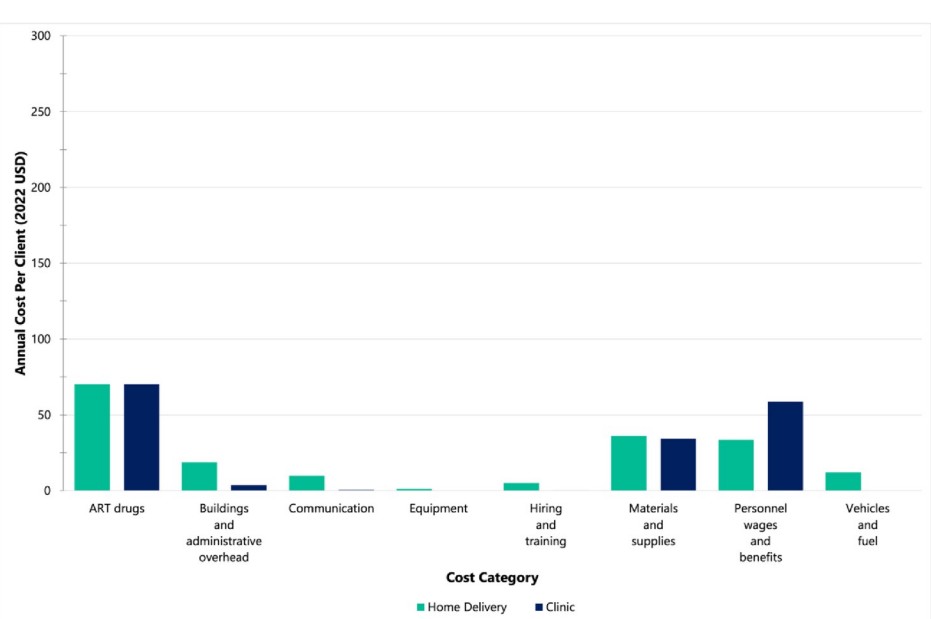

**B**

**Fig 2. At-scale average annual cost per client (2022 USD) for home-delivered 3-month ART refills vs. clinic-based 3-month ART refills (standard of care) by cost category in the programmatic NDoH-implemented scenario.** The NDoH scenario assumes fixed costs as implemented in the Deliver Health Study and public sector clinical staff salaries instead of study salaries. (A) Home-delivered ART intervention (first year costs) vs. clinic-based ART refills. (B) Home-delivered ART intervention (subsequent year costs) vs. clinic-based ART refills.

in observed incremental costs given prior literature from South Africa, which suggests that the cost of delivering ART is on average 7% more costly in rural areas than in urban areas [6]. Second, our estimates for the SOC using KwaZulu-Natal's provincial budget data ($167 per client

annually) are substantially lower than the estimates from the South African HIV investment case, which estimated that the average cost (in 2022 USD) for providing ART per client annually across South Africa was $290 [37]–a potential explanation for our higher estimated cost of SOC compared with the HIV investment case is that our study represents KwaZulu-Natal province while the HIV investment case is a country-wide average. Our estimated incremental cost of home-delivered ART would be lower and appear even less expensive if our SOC estimate reflected a rural setting or used the South Africa HIV investment case–adding validity to our interpretation that the incremental cost is modest. Surprisingly, our estimated at-scale annual per-client costs of ART with quarterly refills were lower than those observed in the DO ART Study, a study of community-based ART conducted in the same region of rural South Africa ($183 in this study, compared with $287 in the DO ART Study for subsequent years after adjusting for inflation as shown in S2 File) [33]–this likely reflects the reasonably low vehicle and fuel costs required for home delivery of ART. The DO ART follow-on longitudinal modelling study found that community-based ART cost-effectively improved HIV clinical outcomes among men and accordingly prevented new infections and improved outcomes among women [38]. Since the Deliver Health Study found that home-based ART delivery also increased viral suppression overall and among men with lower observed costs than in DO ART [15], we might anticipate that home delivery is also cost-effective, though a comprehensive economic evaluation is needed. Finally, in line with other studies on multi-month scripting from Southern Africa which have also found comparable clinical outcomes [39–42], we found that increasing the duration of ART scripts to six- and 12-months further reduced costs due to the fewer follow-up visits.

Our study had a number of limitations. First, as discussed above, our costing of home-delivered ART covered a rural setting while the SOC covered a rural and urban setting, likely leading to an overestimate of incremental costs [43]. In a related point, we did not obtain all costing inputs from a single source; however, we relied on either peer-reviewed studies or published data from the South African government. Third, our study included a low client volume from a pilot randomized controlled trial leading to higher costs per person. To account for the small sample size, we modelled at-scale scenarios to accommodate a higher client volume; these findings were more in line with estimates from the literature. Fourth, we estimated costs for TDF/FTC/EFV (the regimen prescribed for Deliver Health Study participants); however, as of 2023, South Africa recommends ART regimens containing DTG for greater adherence and viral suppression [17, 44–46]–future costing studies should note this change in clinical guidelines. Fifth, while our study was conducted from the provider perspective to reflect the stakeholder of interest, a societal perspective could better capture benefits to clients (e.g., transportation costs, lost wages due to time spent on clinic visits, childcare [11]). Additionally, since the provider was the South African NDoH, we did not consider the costs if the intervention had been implemented in the private sector. Sixth, our estimates assume that clients do not relocate. If clients moved to a further destination but wanted to continue the home delivery service, then transportation costs incurred by the intervention payer would likely increase. Seventh, this study uses the SOC as the comparison, but does not compare to costs of other DSD models for HIV, such as the use of community package lockers (smart lockers) being tested in an ongoing trial throughout rural KwaZulu-Natal [47]. Finally, while the present study does not incorporate health outcomes, the estimates generated in this study can be leveraged for future cost-effectiveness analysis.

National scale-up of DSD models for HIV treatment, tailored to specific patient populations and geographies, are needed in South Africa. A number of alternative ART dispensing mechanisms have demonstrated improved medication adherence and retention compared to clinic care, including home delivery service, community package lockers, and automated pharmacy

dispensing units [48]. These methods, particularly when clients can flexibly choose the most convenient option, can strengthen individuals' commitment to care [49, 50]. Additionally, community-based ART resupply can increase coverage in rural areas [51] and reduce clinic congestion [52]. At-scale, home-delivered ART could be optimized using a centralized warehouse to fulfill and dispatch delivery trucks that synchronize deliveries to neighboring locations, similar to what is used in the private sector by Amazon [53]. These warehouses could even be used for other chronic medications as part of the Centralised Chronic Medicine Dispensing and Distribution [54], freeing up clinics and pharmacies to focus on clients with health care needs other than refills. Drone delivery, which has successfully been tested in other African countries including Uganda (for ART) and Malawi (for transportation of lab samples), offers a promising cost-effective solution for rural areas [55, 56]. In fact, Amazon has recently begun implementing home delivery of medications via drones in the US [57]. These innovative methods hold promise for addressing the hard-to-reach and high prevalence populations in rural South Africa, where interventions increasing early adoption of and adherence to ART can be highly cost-effective (increases cost-effectiveness relative to the base-case by 23% in a rural areas of South Africa, compared to 1.1% in urban areas) [6]. Future research is warranted to observe whether 12 home deliveries a day per team is indeed feasible at-scale, how costs change in scale-up (particularly with regard to shifts in fixed and variable costs), how actual observation of intervention implementation would account for real-life inefficiencies and therefore be more accurate, and to explore the costs and outcomes of various innovative approaches compared to traditional clinic-based care.

## Conclusion

In this costing study of home-delivered ART and monitoring in rural KwaZulu-Natal, South Africa, we found that within the context of a low-volume pilot study, home-delivered ART and monitoring incurred higher annual per-client costs to the health system compared to standard three-month clinic-based ART refills ($714 compared with $167 after the first year), primarily driven by personnel costs. However, our analysis suggests that scaling up the home delivery program to 12 home deliveries per day would make the per-client cost comparable to the standard of care ($183 compared with $167 after the first year). Moreover, the incremental cost of home delivery could be further reduced–and even become cost-saving–by using six- or 12-month refill scripts ($177 and $135, respectively). The findings of this study could help inform the South African NDoH's strategy to meet the 95-95-95 targets by focusing on the scale-up of novel DSD methods, such as home-delivered ART, to promote long-term retention in HIV care while reducing patient load at public health clinics.

## Supporting information

**S1 File. Narrative summary of the Deliver Health Study.**
(DOCX)

**S2 File. Additional methods.**
(DOCX)

**S3 File. Additional results.**
(DOCX)

## Acknowledgments

We would like to thank the Deliver Health Study Team and the individuals who participated in the study.

## Author Contributions

**Conceptualization:** Ashley S. Tseng, Ruanne V. Barnabas, Alastair van Heerden, Maitreyi Sahu.

**Data curation:** Ashley S. Tseng, Xolani Ntinga.

**Formal analysis:** Ashley S. Tseng.

**Funding acquisition:** Ashley S. Tseng, Ruanne V. Barnabas, Alastair van Heerden.

**Investigation:** Ashley S. Tseng, Ruanne V. Barnabas, Alastair van Heerden, Maitreyi Sahu.

**Methodology:** Ashley S. Tseng, Ruanne V. Barnabas, Maitreyi Sahu.

**Project administration:** Ashley S. Tseng, Ruanne V. Barnabas, Alastair van Heerden, Xolani Ntinga, Maitreyi Sahu.

**Supervision:** Ruanne V. Barnabas, Maitreyi Sahu.

**Validation:** Xolani Ntinga, Maitreyi Sahu.

**Visualization:** Ashley S. Tseng.

**Writing – original draft:** Ashley S. Tseng.

**Writing – review & editing:** Ashley S. Tseng, Ruanne V. Barnabas, Alastair van Heerden, Xolani Ntinga, Maitreyi Sahu.

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
