## [Decision Letter · Decision Letter 0]

28 Aug 2024

PGPH-D-24-01215

Costs of home-delivered antiretroviral therapy refills for persons living with HIV: evidence from a pilot randomized controlled trial in KwaZulu-Natal, South Africa

Dear Dr. Ashley Tseng,

Thank you for submitting your manuscript to PLOS Global Public Health. After careful consideration, we feel that it has merit but does not fully meet PLOS Global Public Health’s publication criteria as it currently stands. Therefore, we invite you to submit a revised version of the manuscript that addresses the points raised during the review process.

We look forward to receiving your revised manuscript.

Kind regards,

Henry Zakumumpa, PhD

Academic Editor

Journal Requirements:

Additional Editor Comments (if provided):

We are delighted to share the feedback from our reviewers. One of the reviewers has requested more clarity around the methods of this study which deserves your attention. Please provide a point-by-point response to the comments raised and indicated the changes made or not (with justification). This will enable us to move swiftly to a decision.

Reviewers' comments:

Reviewer's Responses to Questions

**Comments to the Author**

1. Does this manuscript meet PLOS Global Public Health’s publication criteria? Is the manuscript technically sound, and do the data support the conclusions? The manuscript must describe methodologically and ethically rigorous research with conclusions that are appropriately drawn based on the data presented.

Reviewer #1: Yes

Reviewer #2: Yes

Reviewer #3: Yes

2. Has the statistical analysis been performed appropriately and rigorously?

Reviewer #1: Yes

Reviewer #2: I don't know

Reviewer #3: Yes

3. Have the authors made all data underlying the findings in their manuscript fully available (please refer to the Data Availability Statement at the start of the manuscript PDF file)?

Reviewer #1: No

Reviewer #2: Yes

Reviewer #3: Yes

4. Is the manuscript presented in an intelligible fashion and written in standard English?

Reviewer #1: Yes

Reviewer #2: Yes

Reviewer #3: Yes

5. Review Comments to the Author

Reviewer #1: The paper is well written especially highlighting the importance of home-delivered antiretroviral therapy refills in enhancing accessibility to HIV care in rural areas. I note however, the authors do not readily avail the data unless when requested through writing.

Reviewer #2: The costing perspective for this study is the provider's perspective. Using the term "payers" perspective is erroneous because the payer could be the provider or the client/services consumer.

It is not well explained why the home-based care has costs such as buildings and equipment. These would be expected at health facility level only. A better explanation needs to come out clearly.

What is the basis for assuming 12 deliveries per day at scale?

Why didn't the authors estimate the cost of ART delivery for DTG (and compare it with TDF/FTC/EFV?

Reviewer #3: This study is well-executed, with clear articulation of the objectives, methods, and results. The discussion section effectively interprets the evidence and thoughtfully addresses the study's limitations. The authors highlight that while the unit costs for the observed home-delivery program in the study were initially high, these costs can be significantly reduced through scale-up with a higher patient volume and by increasing the duration of ART refills. As a result, the costs become comparable to those of clinic-based ART delivery. The findings of this study provide valuable insights that will be highly useful for national strategy cost-projections and for researchers exploring the costs and cost-effectiveness of innovative ART delivery methods.

Additional comments for authors:

• Introduction:

o Line 74- ‘successful’ treatment is ambiguous. Based on context, you’re likely referring to the cost of achieving viral suppression- so say that: “cost of achieving viral suppression is higher due to xxx”, but if you mean enrollment and retention to care –specify that.

o Paragraph 1: You’ve effectively highlighted the greater prevalence and higher costs associated with HIV treatment in rural areas. To further reinforce your argument, can you include any statistics on ART coverage in rural vs. urban areas? Demonstrating lower ART coverage in rural areas would underscore inequities and thus help argue the need for home-based ART delivery in those settings.

• Methods

o Considering that vehicle and fuel costs represented one of the largest cost categories- please specify in your methods what type of vehicle costs are included: Are these costs related to a single car, multiple motorcycles, recurring reimbursements for public transport, or a combination of these? Providing this detail would be useful for those looking to replicate the study.

o Lines 142-151: please specify the time period for which costs were retrospectively collected for both study arms.

• Results:

o Not vital, but if you’d like you could add data labels to your figures, including the % of total costs per arm.

• Discussion:

o Lines 323-324: Differences in costs with HIV investment case- of $290 vs. $167- are significantly different—and that warrants an explanation. Without an explanation, readers might be left questioning the validity/applicability of findings. You might consider geo/demographic differences, differences in methods, perhaps costs of certain inputs have changed, or scope (ie perhaps some other costs have been included/excluded).

o Lines 327-332: Confusing. The first part of the sentence states that costs in your study ($183) were lower than those observed in the DO ART Study ($287). However, the next sentence implies that home delivery should be more expensive than community-based delivery. This creates a contradiction: the data presented suggests that home delivery was actually less expensive, but the explanation given suggests the opposite should be true. You should keep the insights you have written, saying that you expected inverse results because fuel/time, but [provide possible reasons for lower costs].

o Line 333: remove second ‘that’

o Your methods for estimating scale-up costs are well reasoned, and you justify the 12 home deliveries/day as a reasonable amount. The estimation is a good starting point, but it would be important to acknowledge in discussion that this assumption should be tested in practice. Specifically, further research is warranted to observe whether 12 a day is indeed feasible at scale, how costs change in scale up – particularly with regard to shifts in fixed and variable costs- and how actual observation would account for real-life in/efficiencies and therefore be more accurate.

6. PLOS authors have the option to publish the peer review history of their article (what does this mean?). If published, this will include your full peer review and any attached files.

**Do you want your identity to be public for this peer review?** For information about this choice, including consent withdrawal, please see our Privacy Policy.

Reviewer #1: **Yes: **Richard Ssempala

Reviewer #2: No

Reviewer #3: No

---

## [Decision Letter · Decision Letter 1]

11 Oct 2024

Costs of home-delivered antiretroviral therapy refills for persons living with HIV: evidence from a pilot randomized controlled trial in KwaZulu-Natal, South Africa

PGPH-D-24-01215R1

Dear Ashley S. Tseng,

We are pleased to inform you that your manuscript 'Costs of home-delivered antiretroviral therapy refills for persons living with HIV: evidence from a pilot randomized controlled trial in KwaZulu-Natal, South Africa' has been provisionally accepted for publication in PLOS Global Public Health.

Best regards,

Henry Zakumumpa, PhD

Academic Editor

Reviewer Comments (if any, and for reference):

Reviewer's Responses to Questions

**Comments to the Author**

1. If the authors have adequately addressed your comments raised in a previous round of review and you feel that this manuscript is now acceptable for publication, you may indicate that here to bypass the “Comments to the Author” section, enter your conflict of interest statement in the “Confidential to Editor” section, and submit your "Accept" recommendation.

Reviewer #1: All comments have been addressed

2. Does this manuscript meet PLOS Global Public Health’s publication criteria? Is the manuscript technically sound, and do the data support the conclusions? The manuscript must describe methodologically and ethically rigorous research with conclusions that are appropriately drawn based on the data presented.

Reviewer #1: Yes

3. Has the statistical analysis been performed appropriately and rigorously?

Reviewer #1: Yes

4. Have the authors made all data underlying the findings in their manuscript fully available (please refer to the Data Availability Statement at the start of the manuscript PDF file)?

Reviewer #1: Yes

5. Is the manuscript presented in an intelligible fashion and written in standard English?

Reviewer #1: Yes

6. Review Comments to the Author

Reviewer #1: The author have clearly responded to my question, where needed they updated the draft to meet my concerns.

7. PLOS authors have the option to publish the peer review history of their article (what does this mean?). If published, this will include your full peer review and any attached files.

**Do you want your identity to be public for this peer review?** For information about this choice, including consent withdrawal, please see our Privacy Policy.

Reviewer #1: **Yes: **Richard Ssempala- Economic Theory and Analysis, Makerere University School of Economics, Kampala, Uganda
